# Adversarial Fisher Vectors for Unsupervised Representation Learning

**Shuangfei Zhai   Walter Talbott   Carlos Guestrin   Joshua M. Susskind**
Apple Inc.
{szhai, wtalbott, guestrin, jsusskind}@apple.com

## Abstract

We examine Generative Adversarial Networks (GANs) through the lens of deep Energy Based Models (EBMs), with the goal of exploiting the density model that follows from this formulation. In contrast to a traditional view where the discriminator learns a constant function when reaching convergence, here we show that it can provide useful information for downstream tasks, e.g., feature extraction for classification. To be concrete, in the EBM formulation, the discriminator learns an unnormalized density function (i.e., the negative energy term) that characterizes the data manifold. We propose to evaluate both the generator and the discriminator by deriving corresponding Fisher Score and Fisher Information from the EBM. We show that by assuming that the generated examples form an estimate of the learned density, both the Fisher Information and the normalized Fisher Vectors are easy to compute. We also show that we are able to derive a distance metric between examples and between sets of examples. We conduct experiments showing that the GAN-induced Fisher Vectors demonstrate competitive performance as unsupervised feature extractors for classification and perceptual similarity tasks. Code is available at `https://github.com/apple/ml-afv`.

## 1   Introduction

Generative adversarial networks (GANs) [1] are state-of-the-art generative modeling methods, where a discriminator network is jointly trained with a generator network to solve a minimax game. According to the original theory in [1], the discriminator reduces to a constant function that assigns a score of $0.5$ everywhere when Nash Equilibrium is reached, making the discriminator useless for anything beyond training the generator. Moreover, the generator models the data density, but in an implicit form that precludes its application to scenarios where an explicit density estimate would be useful. Recently, [2, 3, 4] show that training an energy-based model (EBM) with a parameterized variational distribution reduces to a similar minimax GAN game. This EBM view, in contrast to the original GAN formulation, leads to an interpretation where the discriminator itself is an explicit density model of the data.

We show that under certain approximations, the deep EBMs can be trained with a modern GAN implementation (see Sec. 2.2). We then focus on exploring the utility of the density models learned according to the EBM interpretation. Inspired by the approach in [5], we show how to use the Fisher Score and Fisher Information induced by the learned density model to compute representations of data samples. Namely, we derive normalized Fisher Vectors and Fisher Distance measure to estimate similarities both between individual data samples and between sets of samples. We call these derived representations Adversarial Fisher Vectors (AFVs).

We propose to apply the density model and derived AFV representation in several ways. First, we show that the learned AFV representations are useful as pre-trained features for linear classification tasks, and that the similarity function induced by the learned density model can be used as a perceptual

metric that correlates well with human judgments. Second, we show that estimating similarities between sets using AFV allows us to monitor the training process. Notably, we show that the Fisher Distance between the set of validation examples and generated examples can effectively capture the notion of overfitting, which is verified by the quality of the corresponding AFVs.

As an additional benefit of the EBM interpretation, we provide a view of the generator update as an approximation to stochastic gradient Markov Chain Monte Carlo (MCMC) sampling [6], similar to [4]. We show that directly enforcing local updates of the generated examples improves training stability, especially early in training.

By fully utilizing the density model derived from the EBM framework, we make the following contributions:

- AFV representations are derived for unsupervised feature extraction and similarity estimation from the learned density model.
- GAN training is improved through monitoring (AFV metrics) and stability (MCMC updates).
- AFV is shown to be useful for extracting unsupervised features, leading to state-of-the-art performance on classification and perceptual similarity benchmarks.

## 2 Background

### 2.1 Generative Adversarial Networks

GANs [1] learn a discriminator and generator network simultaneously by solving a minimax game:

$$\max_G \min_D E_{\mathbf{x} \sim p_{data}(\mathbf{x})}[-\log D(\mathbf{x})] - E_{\mathbf{z} \sim p_{\mathbf{z}}(\mathbf{z})}[\log(1 - D(G(\mathbf{z})))], \tag{1}$$

where $p_{data}(\mathbf{x})$ denotes the data distribution; $D(\mathbf{x})$ is the discriminator that takes as input a sample and outputs a scalar in $[0, 1]$; $G(\mathbf{z})$ is the generator that maps a random vector $\mathbf{z} \in R^d$ drawn from a pre-defined distribution $p(\mathbf{z})$. Equation 1 suggests a training procedure consisting of two loops: in the inner loop $D$ is trained until convergence given $G$, and in the outer loop $G$ is updated one step given D. [1] shows that GANs implicitly minimize the Jensen-Shannon divergence between the generator distribution $p_G(\mathbf{x})$ and the data distribution $p_{data}(\mathbf{x})$, and hence samples from $G$ approximate the true data distribution when the minimax game reaches Nash Equilibrium.

### 2.2 GANs as variational training of deep EBMs

Following [3], let an EBM define a density function as: $p_E(\mathbf{x}) = \frac{e^{-E(\mathbf{x})}}{\int_{\mathbf{x}} e^{-E(\mathbf{x})} d\mathbf{x}}$, where $E(\mathbf{x})$ is the energy of input $\mathbf{x}$. We can then write its negative log likelihood (NLL) as: $E_{\mathbf{x} \sim p_{data}(\mathbf{x})}[E(\mathbf{x})] + \log[\int_{\mathbf{x}} e^{-E(\mathbf{x})} d\mathbf{x}]$, which can be further developed as:

$$E_{\mathbf{x} \sim p_{data}(\mathbf{x})}[E(\mathbf{x})] + \log \int_{\mathbf{x}} q(\mathbf{x}) \frac{e^{-E(\mathbf{x})}}{q(\mathbf{x})} d\mathbf{x} = E_{\mathbf{x} \sim p_{data}(\mathbf{x})}[E(\mathbf{x})] + \log E_{\mathbf{x} \sim q(\mathbf{x})}[\frac{e^{-E(\mathbf{x})}}{q(\mathbf{x})}]$$

$$\geq E_{\mathbf{x} \sim p_{data}(\mathbf{x})}[E(\mathbf{x})] + E_{\mathbf{x} \sim q(\mathbf{x})}[\log \frac{e^{-E(\mathbf{x})}}{q(\mathbf{x})}] = E_{\mathbf{x} \sim p_{data}(\mathbf{x})}[E(\mathbf{x})] - E_{\mathbf{x} \sim q(\mathbf{x})}[E(\mathbf{x})] + H(q), \tag{2}$$

where $q(\mathbf{x})$ is an auxiliary distribution which we call the *variational distribution*, with $H(q)$ denoting its entropy. Equation 2 is a straightforward application of Jensen's inequality, and it gives a variational lower bound on the NLL given $q(\mathbf{x})$. The lower bound is tight when $\frac{e^{-E(\mathbf{x})}}{q(\mathbf{x})}$ is a constant w.r.t. $\mathbf{x}$, i.e., $q(\mathbf{x}) \propto e^{-E(\mathbf{x})}$, $\forall \mathbf{x}$, which implies that $q(\mathbf{x}) = p_E(\mathbf{x})$. We then let $D(\mathbf{x}) = -E(\mathbf{x})$ and $q(\mathbf{x}) = p_G(\mathbf{x})$ (i.e., the implicit distribution defined by a generator as in GANs), which leads to an objective as follows:

$$\min_D \max_G E_{\mathbf{x} \sim p_{data}(\mathbf{x})}[-D(\mathbf{x})] + E_{\mathbf{z} \sim p_{\mathbf{z}}(\mathbf{z})}[D(G(\mathbf{z}))] + H(p_G), \tag{3}$$

where in the inner loop, the variational lower bound is maximized w.r.t. $p_G$; the energy model then is updated one step to decrease the NLL with the optimal $p_G$ (see Figure 1).

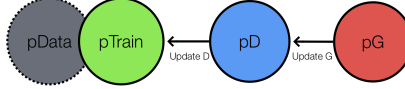

Figure 1: EBM view of GAN, in which the generator is first updated such that sampling from $P_G(\mathbf{x})$ approximates the discriminator distribution $P_D(\mathbf{x})$; the discriminator is then updated to fit $P_D(\mathbf{x})$ to $P_{data}(\mathbf{x})$.

Equation 3 and Equation 1 bear a lot of similarity, both taking the form of a minimax game between $D$ and $G$. The three notable differences, however, are 1) the emergence of the entropy regularization term $H(p_G)$ in Equation 3 which in theory prevents the generator from collapsing[1], 2) the order of optimizing $D$ and $G$, and 3) $D$ is a density model for the data distribution and $G$ learns to sample from $D$.

In practice, it is difficult to come up with a differentiable approximation to the entropy term $H(p_G)$. We instead rely on implicit regularization methods such as using Batch Normalization [8] on the generator (see Sec. 3.2 for more analysis). This simplification makes it possible to implement Equation 3 in exactly the same way as a GAN, where $D$ and $G$ are alternately updated on a few mini-batches of data. We can then borrow the implementation of a state-of-the-art GAN, and focus on utilizing the trained model as discussed below.

## 3    Methodology

### 3.1    Adversarial Fisher Vectors

In the EBM view of GANs, the discriminator is a dual form of the generator, where in the perfect scenario each defines a distribution that matches the training data. By nature, interpreting the generator distribution can be easily done by first sampling from it, and inspecting the quality of samples produced. However, it is not clear how to evaluate or use a discriminator, even under the assumption that it captures as much information about the training data as the generator does. To this end, we turn our eye to the theory of Fisher Information [9, 5], starting by adopting the EBM view of GANs as discussed before.

Given a density model $p_\theta(\mathbf{x})$, with $\mathbf{x} \in R^d$ as the input and $\theta$ being the model parameters, we can derive the Fisher Score of an example $\mathbf{x}$ is as $U_\mathbf{x} = \nabla_\theta \log p_\theta(\mathbf{x})$. Intuitively, the Fisher Score encodes the desired change of model parameters to better fit the example $\mathbf{x}$. We further define the Fisher Information as $\mathcal{I} = \mathrm{E}_{\mathbf{x} \sim p_\theta(\mathbf{x})}[U_\mathbf{x} U_\mathbf{x}^T]$. According to information geometry theory [10], the generative model defines a local Riemannian manifold, over model parameters $\theta$, with a local metric given by the Fisher Information. Following [5], one can then use the Fisher Score to map an example $\mathbf{x}$ to the model space, and measure the proximity between two examples $\mathbf{x}, \mathbf{y}$ by $U_\mathbf{x}^T \mathcal{I}^{-1} U_\mathbf{y}$. One can also naturally adopt the same principle to induce a distance metric as $\mathcal{D}(\mathbf{x}, \mathbf{y}) = (U_\mathbf{x} - U_\mathbf{y})^T \mathcal{I}^{-1}(U_\mathbf{x} - U_\mathbf{y})$, which we call the Fisher Distance. Additionally, we can generalize the notion of Fisher Distance to take two sets of examples as input:

$$\mathcal{D}(\mathbf{X}, \mathbf{Y}) = (\frac{1}{|\mathbf{X}|}\sum_{\mathbf{x} \in \mathbf{X}} U_\mathbf{x} - \frac{1}{|\mathbf{Y}|}\sum_{\mathbf{y} \in \mathbf{Y}} U_\mathbf{y})^T \mathcal{I}^{-1}(\frac{1}{|\mathbf{X}|}\sum_{\mathbf{x} \in \mathbf{X}} U_\mathbf{x} - \frac{1}{|\mathbf{Y}|}\sum_{\mathbf{y} \in \mathbf{Y}} U_\mathbf{y}).$$

We finally define the AFV of an example as:

$$V_\mathbf{x} = \mathcal{I}^{-\frac{1}{2}} U_\mathbf{x},$$

so the Fisher Distance is equivalent to the Euclidean distance with AFVs.

AFVs provide a valuable tool for utilizing a generative model. Given a fixed model, two examples are considered identical only if the desired change to the model parameters are the same. As a simple illustration, for a standard Gaussian distribution $\mathcal{N}(\mu, \sigma)$, two examples $x_1 = \mu + 1$ and $x_2 = \mu - 1$ have exactly the same density, but still have different Fisher Scores. As a matter of fact, classical

Fisher Vectors have successfully been applied by utilizing relatively simple density models such as Mixture of Gaussians, see [11] for detailed examples.

In the context of EBMs, we parameterize the density model in intervals of $D$ as $p_\theta(\mathbf{x}) = \frac{e^{D(\mathbf{x};\theta)}}{\int_{\mathbf{x}} e^{D(\mathbf{x};\theta)} d\mathbf{x}}$ with $\theta$ explicitly being the parameters of $D$. We then derive the Fisher Score as

$$U_{\mathbf{x}} = \nabla_\theta D(\mathbf{x};\theta) - \nabla_\theta \log \int_{\mathbf{x}} e^{D(\mathbf{x};\theta)} d\mathbf{x} = \nabla_\theta D(\mathbf{x};\theta) - \mathrm{E}_{\mathbf{x} \sim p_\theta(\mathbf{x})} \nabla_\theta D(\mathbf{x};\theta). \qquad (4)$$

According to Equation 3, the EBM interpretation of a GAN entails that during training, the generator is updated to match the distribution $p_G(\mathbf{x})$ to $p_\theta(\mathbf{x})$. This allows us to conveniently approximate the second term by sampling from the generator's distribution, resulting in the Fisher Score and Fisher Information we work with:

$$U_{\mathbf{x}} = \nabla_\theta D(\mathbf{x};\theta) - \mathrm{E}_{\mathbf{z} \sim p(\mathbf{z})} \nabla_\theta D(G(\mathbf{z});\theta), \ \mathcal{I} = \mathrm{E}_{\mathbf{z} \sim p(\mathbf{z})}[U_{G(\mathbf{z})} U_{G(\mathbf{z})}^T]. \qquad (5)$$

By approximating the density model defined by $D$ with the learned generator distribution, we have come up with a scalable approximation to the Fisher Score and Fisher Information for an un-normalized deep density model. In particular, the Fisher Score takes the form of the gradient of the discriminator (the negative energy in EBM terms) output w.r.t. its parameters, subtracted by the average gradient of all generated examples. The Fisher Information, on the other hand, elegantly reduces to the covariance matrix of the gradient of the generated examples.

For practical settings where $D$ takes the form of a deep convolutional neural network, directly computing the AFVs can be expensive, as the vectors can easily be up to millions of dimensions. We thus resort to a diagonal approximation of the Fisher Information, which yields an efficient form of the AFV:

$$V_{\mathbf{x}} = (diag(\mathcal{I})^{-\frac{1}{2}}) U_{\mathbf{x}}, \qquad (6)$$

where $\mathcal{I}$ and $U_{\mathbf{x}}$ are as defined in Equation 5, and $diag$ denotes the diagonal matrix operator.

## 3.2 Generator update as stochastic gradient MCMC

The use of a generator provides an efficient way of drawing samples from the EBM. However, in practice, great care needs to be taken to make sure that $G$ is well conditioned to produce examples that cover enough modes of $D$. There is also a related issue where the parameters of $G$ will occasionally undergo sudden changes, generating samples drastically different from iteration to iteration, which contributes to training instability and lower model quality.

In light of these issues, we provide a different treatment of $G$, borrowing inspirations from the Markov Chain Monte Carlo (MCMC) literature. MCMC variants have been widely studied in the context of EBMs, which can be used to sample from an unnormalized density and approximate the partition function [12, 13]. Stochastic gradient MCMC [6] is of particular interest as it utilizes the gradient of the log probability w.r.t. the input, and performs gradient ascent to incrementally update the samples (while adding noise to the gradients). See [14] for a recent application of this technique to deep EBMs. We speculate that it is possible to train $G$ to mimic the the stochastic gradient MCMC update rule, such that the samples produced by $G$ will approximate the true model distribution.

To be concrete, we want to constrain the $G$ updates to be local w.r.t. the generated examples, similar to one step stochastic gradient MCMC sampling. To do this, We maintain an old copy of $G$ denoted as $\bar{G}$ (e.g., obtained by Polyak averaging of $G$ parameters) and let the $G$ objective be: $\min_G \mathrm{E}_{\mathbf{z} \sim p(\mathbf{z})}[\frac{1}{2}\|G(\mathbf{z}) - \bar{G}(\mathbf{z}) + \lambda\epsilon\|^2 - \frac{\lambda}{2} D(G(\mathbf{z}))]$. Here $\lambda \in R^+$ is a small scalar quantity that corresponds to the step size of the stochastic gradient MCMC update, $\epsilon \sim \mathcal{N}(0, \mathcal{I})$ is white Gaussian noise. It is not hard to see that the non-parametric local minimum for Equation 7 w.r.t. $G(\mathbf{z})$ is $G(\mathbf{z}) = \bar{G}(\mathbf{z}) + \frac{\lambda}{2} \nabla_{\mathbf{x}} D(\mathbf{x})|_{\mathbf{x} = \bar{G}(\mathbf{z})} + \lambda\epsilon$, which corresponds to one step of stochastic gradient MCMC update, where the starting point is given by $\bar{G}(\mathbf{z})$. In practice, we can optionally ignore the $\epsilon$ term; we can also adopt the same principle to any loss variants such as hinge loss [15], or squared loss, which gives us a generalized form of the MCMC objective:

$$\min_G \mathrm{E}_{\mathbf{z} \sim p(\mathbf{z})}[L(D(G(\mathbf{z}))) + \gamma\|G(\mathbf{z}) - \bar{G}(\mathbf{z})\|^2], \qquad (7)$$

where $L$ denotes a loss function and $\gamma \in R^+$ is a weight factor. Note that this explicit regularization is not always necessary in the sense that the local update can also be implicitly achieved via careful

architecture design and training scheduling. However, we have found that including this term can effectively reduce the mode collapse problem and increase training stability, especially early in training where $G$ undergoes large gradients. This MCMC inspired objective is also similar to the historical averaging trick proposed in [7], but with the importance distinction that we constrain the locality of updates in the sample space, rather than the parameter space.

## 4 Related Work

A number of GAN variants have been proposed by extending the notion of discriminator to a critic that measures the discrepancy of two distributions, with notable examples including Wasserstein GAN [16], f-GAN [17], MMD-GAN [18]. Of particular interest to this work is the connection between GANs and deep energy-based models (EBMs) [19]. It is shown that the training procedure of a GAN resembles that of a deep EBM with variational inference [2, 3, 4, 15]. Our work differs from these in the sense that we directly utilize the discriminator by taking advantage of the fact that it learns a density model of data. There has also been an increasing in the interest of deep EBMs trained with traditional sampling approaches, see [20, 21]. Our implementation of EBMs has the benefit of directly learning a parameterizaed sampler, which is more efficient than iterative MCMC based sampling approaches.

Other works have introduced techniques to improve GAN stability using regularization techniques such as adding noise [22], gradient penalization [23], or conditioning the weights with spectral normalization [24]. Mode collapse has been tackled by encouraging the model to generate high entropy samples [7] and by introducing new training formulations [16]. These techniques are typically adhoc and lack a formal justification. We show a particular connection of our MCMC based $G$ update rule to the the gradient penalty line of work as in [23, 25]. To see this, instead of always sampling from the generator, we allow a small probability $\rho$ to sample particles starting from a real example $\mathbf{x}$. Plugging this in the $D$ objective, we obtain:

$$\min_D \mathrm{E}_{\mathbf{x} \sim p_{data}(\mathbf{x})}[-D(\mathbf{x})] + (1-\rho)\mathrm{E}_{\mathbf{z} \sim p_{\mathbf{z}}(\mathbf{z})}[D(G(\mathbf{z}))]-$$

$$\rho \mathrm{E}_{\mathbf{x} \sim p_{data}(\mathbf{x})}[-D(\mathbf{x} + \frac{\lambda}{2}\nabla_{\mathbf{x}}D(\mathbf{x}) + \lambda\epsilon)]$$

$$\approx \min_D \mathrm{E}_{\mathbf{x} \sim p_{data}(\mathbf{x})}[-(1-\rho)D(\mathbf{x})] + (1-\rho)\mathrm{E}_{\mathbf{z} \sim p_{\mathbf{z}}(\mathbf{z})}[D(G(\mathbf{z}))]+$$

$$\frac{\rho\lambda}{2}\mathrm{E}_{\mathbf{x} \sim p_{data}(\mathbf{x})}\|\nabla_{\mathbf{x}}D(\mathbf{x})\|^2,$$

which is exactly the zero centered gradient penalty regularization proposed in [25].

In early work incorporating generative models into discriminative classifiers, [5] showed that one can use Fisher Information to derive a measure of the similarity between examples. [11] extended this work by introducing the Fisher Vector representation for image classification using Gaussian mixture models for density modeling. More recently, Fisher Information has also been applied to tasks such as meta learning [26]. In this work, we show that it is possible to extend this idea to state-of-the-art deep generative models. In particular, by utilizing the generator as a learned sampler from the density model, we are able to overcome the difficulty of computing the Fisher Information from an un-normalized density model. Compared with other unsupervised representation learning approaches, such as VAEs [27], BiGAN [28], AVFs do not need to learn an explicit encoder. Compared with self-supervised learning approaches, such as [29, 30, 31, 32], our approach is density estimation based, and do not need any domain specific priors to create self supervision signal.

## 5 Experiments

### 5.1 Setup

We conduct our experiments on images of size $32 \times 32$, using CIFAR10 and CIFAR100 [33], CelebA [34] and ImageNet [35]. Our architecture is a re-implementation of the architecture used in [23], with the addition of Spectral Normalization (SN) [24] to the discriminator weights, and a final Sigmoid

Table 1: Evaluating feature extraction techniques w.r.t. classification accuracy on CIFAR10 with a linear classifier. Here AFV-k-n denotes AFV trained with model channel size k and with n examples. D-pool corresponds to using the pooled features from four layers using the same discriminator as in AFV-128-50000. When using linear classifiers on top of pre-trained features, AFV outperforms state-of-the-art classifiers by a large margin. Remarkably, the classification accuracy increases as we add more data, either in the form of data augmentation or even out of distribution data (CIFAR100) during the GAN training phase.

| Method | CIFAR10 | CIFAR100 | Method | #Features |
|---|---|---|---|---|
| Examplar CNN [29] | 84.3 | - | Unsupervised | - |
| DCGAN [38] | 82.8 | - | Unsupervised | - |
| Deep Infomax [39] | 75.6 | 47.7 | Unsupervised | 1024 |
| RotNet Linear [30] | 81.8 | - | Self-Supervised | $\sim$25K |
| AET Linear [32] | 83.3 | - | Self-Supervised | $\sim$25K |
| D-pool-128-50000 | 65.3 | - | Unsupervised | 1.5M |
| AFV-128-50000 | 86.2 | - | Unsupervised | 1.5M |
| AFV-128-50000 + augment | 87.1 | - | Unsupervised | 1.5M |
| AFV-256-50000 + augment | 88.5 | - | Unsupervised | 5.9M |
| AFV-256-50000 + C100 + augment | **89.1** | **67.8** | Unsupervised | 5.9M |
| D + BN supervised training | 92.7 | 70.3 | Supervised | - |

nonlinearity to the discriminator. We adopt the least squares loss as proposed in LSGAN [36] as the default loss for the discriminator, and use the squared loss version of Equation 7 for the generator, with $\gamma = 0$. Unless otherwise mentioned, the channel size for convolutional layers is 128. All experiments use batch size 64, ADAM optimizer [37] with $\beta_1 = 0$, $\beta_2 = .999$, learning rate for $G = 2 \times 10^{-4}$, and learning rate for $D = 4 \times 10^{-4}$. By default we train our model with a fixed number of iterations (800K) and obtain the last checkpoint for evaluation, unless otherwise mentioned.

## 5.2 Evaluating AFV representations

An appealing property of the EBM view of GAN training is that the discriminator should be able to learn a density function that characterizes the data manifold of the training set. This is in stark contrast to GAN theory, where $D$ reduces to a constant function at convergence. To verify the usefulness of a trained discriminator, we trained a set of models with different settings and compute the AFVs for the dataset. To be concrete, we start from the default architecture and experiment with adding data augmentation, increasing the size of the model by using 256 channels for both $D$ and $G$ and combining CIFAR10 and CIFAR100. We then train a linear classifier with squared hinge loss (L2SVM) on the extracted features to focus on the direct quality of the feature representation as opposed to the power of the classifier. We obtain a state-of-the-art unsupervised pretraining classification accuracy of 89.1% and 67.8 on CIFAR10 and CIFAR100, respectively, as shown in Table 1. These results are also comparable to the supervised learning result with the discriminator's architecture (while replacing Spectral Normalization with Batch Normalization for better performance, shown in the last row). In contrast, a control experiment without data augmentation shows that pooling $D$ features is significantly worse than the extracted AFV representation on CIFAR10 (65.3% vs. 86.2%). In addition, we show in Figure 2 that the AFVs successfully recover a semantically intuitive notion of similarity between classes (e.g., cars are similar to trucks, dogs are similar to cats). Notably, the dimensionality of our AFVs is 3 orders of magnitude higher than those of the existing methods, which would typically bring a higher propensity to overfitting. However, AFVs still show great generalization ability, demonstrating that they are indeed capturing a meaningful low dimensional subspace that allows easy interpolation between examples. See Supplementary Figures S2 and S3 for visualizations of nearest neighbors.

While AFVs capture properties of the data manifold useful for classification and comparing samples, they may contain additional fine-grained perceptual information. Therefore, in our final experiment we examine the usefulness of AFVs as a perceptual similarity metric consistent with human judgments. Following the approach described in [40], we use the AFV representation to compute distances between image patches and compare with existing methods on the Berkeley-Adobe Perceptual Patch Similarity (BAPPS) dataset on 2AFC and Just Noticeable Difference (JND) metrics. We first train a GAN on ImageNet with the same architecture and settings as previous experiments, and then calculate AFVs on the BAPPS evaluation set. Table 2 shows the performance of AFV along with

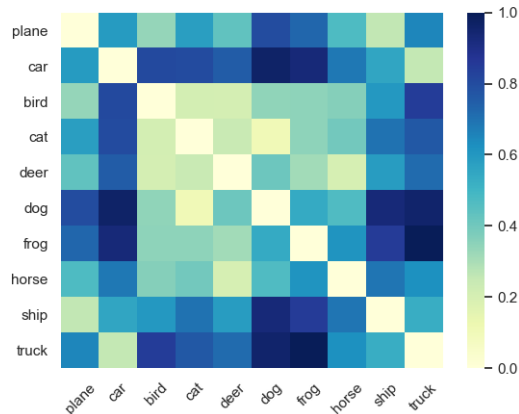

Figure 2: The class distance matrix derived by computing the Fisher Distance between sets of examples representing each class. We see that although trained in an unsupervised way, the Fisher Vectors for each class (set of images) effectively capture semantic similarities between classes.

Table 2: 2AFC and JND scores for different models across traditional and CNN distortion benchmarks reported in [40]. The first three methods are supervised (top). The second two methods are self-supervised (middle). The last two methods, including AFV, are unsupervised (bottom).

| Model | 2AFC (trad) | 2AFC (CNN) | Avg 2AFC | JND (trad) | JND (CNN) | Avg JND |
|---|---|---|---|---|---|---|
| AlexNet [41] | 70.6 | **83.1** | 76.8 | 44.0 | 67.1 | 55.6 |
| SqueezeNet [42] | 73.3 | 82.6 | **78.0** | 49.3 | **67.6** | 58.4 |
| VGG [43] | 70.1 | 81.3 | 75.7 | 47.5 | 67.1 | 57.3 |
| Puzzle [31] | 71.5 | 82.0 | 76.8 | - | - | - |
| BiGAN [28] | 69.8 | 83.0 | 76.4 | - | - | - |
| Stacked K-means [44] | 66.6 | 83.0 | 74.8 | - | - | - |
| AFV (ours) | **74.9** | 80.2 | 77.6 | **55.7** | 62.0 | **58.9** |

a variety of existing benchmarks, averaging across traditional and CNN-based distortion sets as in [40]. AFV exceeds the reported unsupervised and self-supervised methods and is competitive with supervised methods trained on ImageNet. Crucially, AFV is domain-independent, and does not require label-based supervision for training the features or the perceptual similarity metric.

## 5.3 Using the Fisher Distance to monitor training

One of the difficulties of GAN training is the lack of reliable metrics, e.g., a bounded loss function. Recently, domain specific methods, such as Inception Scores (IS) [7] and Fréchet Inception Distance (FID) [45] have been used as surrogate metrics to monitor the quality of generated examples. However, such scores usually rely on a discriminative model trained on ImageNet, and thus have limited applicability to datasets that are drastically different. In this section, we show that monitoring the Fisher Distance between the set of real and generated examples serves as an informative tool to diagnose the training process. To this end, we conducted a set of experiments on CIFAR10 by varying the number of training examples from the set $\{1000, 5000, 25000, 50000\}$. Figure 3 shows the batch-wise estimate of IS and the "Fisher Similarity", which is defined as $e^{-\lambda \mathcal{D}(\mathbf{X}_r, \mathbf{X}_g)}$. Here $\mathbf{X}_r$ and $\mathbf{X}_g$ denotes a batch of real and generated examples, respectively; $\lambda$ is a temperature term which we set as 10.

We see that when the number of training examples is large, the validation Fisher Similarity steadily increases, aligning with Inception Score. On the other hand, when the number of training examples is small, the validation Fisher Similarity starts decreasing at some point. The model then stops learning, where the corresponding Inception Score also saturates. One caveat about using the Fisher Distance is that the score is generally only comparable within the same training run, as the approximation of the Fisher Information is not accurate and the Fisher Information is not invariant under reparameterization.

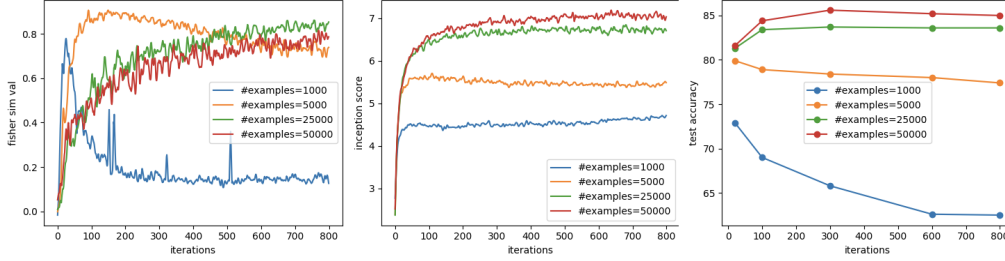

Figure 3: Fisher Similarity of the validation set (left), Inception Scores (middle) by varying the number of training examples from $\{1000, 5000, 25000, 50000\}$, classification test accuracy with AFVs obtained at various checkpoints (right). We see that while the absolute values of the Fisher Similarities are not comparable across models, the trend of the progression of the Fisher Similarities are indicative of optimization and overfitting, which correlates well with the classification accuracy, while the Inception Score fails to do so.

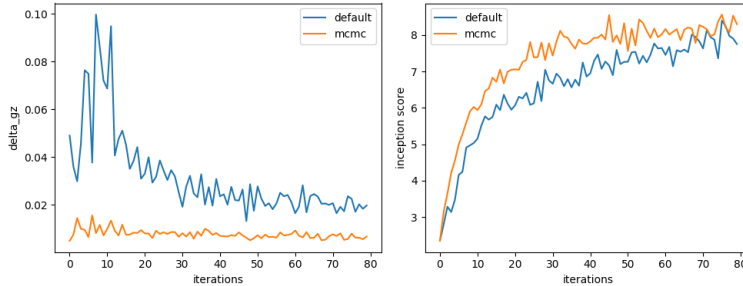

Figure 4: Left: $\Delta G(\mathbf{z})$ of the default generator objective and Equation 7 for the first 80K iterations. Right: the corresponding Inception Scores. In the early phases of training, the MCMC objective maintains small local updates of the generated samples, which results in faster and more stable training.

However, empirically we have found that increasing validation Fisher Similarity is a good indicator of training progress and generalization to unseen data. To support this, we obtained 5 checkpoints from the 4 models at iteration 20, 100, 300, 600, 800, respectively, and trained an L2SVM with the corresponding AFVs, where we show that the classification accuracy correlates well with the validation Fisher Similarity (right panel of Figure 3). The Inception Score does not capture the observed overfitting.

## 5.4 Interpreting $G$ update as parameterized MCMC

One necessary condition for applying AFV is assuming that the generator is approximating the EBM well during the course of training. To test this hypothesis, we trained a model on ImageNet with size $64 \times 64$, and accordingly modify the default architecture by adding one residual block to both the generator and discriminator. We compare the default generator objective and the MCMC objective in Equation 7 in Figure 4, where we show the training statistics in the first $80K$ iterations. We monitor $\Delta(G(\mathbf{z}))$, namely the change of generated examples after one G update, which corresponds to the second term of Equation 7. We see that with the explicit MCMC objective, $\Delta G(\mathbf{z})$ maintains a small quantity always, which also results in an improvement in Inception score over the default objective. Supplementary Figure S4 shows a comparison of generated sample quality when training with and without the MCMC objective. We hypothesize that local updates of $G$ can be achieved via architecture search and learning schedule tuning, but in practice, we have found that using the MCMC objective with a small $\gamma$ often times yield faster training than the standard $G$ losses, especially in the early training phases.

# 6 Conclusion

In this paper, we demonstrated that GANs can be reinterpreted in order to learn representations that have desirable qualities for a diverse set of tasks without requiring domain knowledge or labeled data. We showed that a well trained GAN can capture the intrinsic manifold of data and be used for density estimation following the AFV methodology. We provided empirical analysis supporting the strength of our method. First, we showed that AFVs are a reliable indicator of whether GAN training is well behaved, and that we can use this monitoring to select good model checkpoints. Second, we showed that forcing the generator to track MCMC improves stability and leads to better density models. We next showed that AFVs are a useful feature representation for linear and nearest neighbor classification, achieving state-of-the-art among unsupervised feature representations on CIFAR-10. Finally, we showed that a well-trained GAN discriminator does contain useful information for fine-grained perceptual similarity. Taken together, these experiments show the usefulness of the EBM and associated Fisher Information framework for extracting useful representational features from GANs. In future work, we plan to improve the scalability of the AFV method by compressing the Fisher Vector representation, e.g., using product quantization as in [11].

## Footnotes

[1]Interestingly, similar diversity promoting regularization terms have been independently explored in the GAN literature, such as in [7].

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
