[Supplementary Material]

# Supplementary Material: Adversarial Fisher Vectors for Unsupervised Representation Learning

**Shuangfei Zhai   Walter Talbott   Carlos Guestrin   Joshua M. Susskind**
Apple Inc.
{szhai, wtalbott, guestrin, jsusskind}@apple.com

Figure S1: Progression of various statistics during the course of training a healthy GAN on CIFAR10. From top left to bottom right: training Fisher Similarity, validation Fisher Similarity, Inception Score, training discriminator score, validation discriminator score, generated discriminator score. We see that both the training and validation Fisher Similarity correlates well with Inception Score, healthily increasing. The discriminator's scores on the other hand do not exhibit a clear pattern.

Figure S2: Visualization of 5 nearest training examples of 10 **test** examples using pixel (left), ResNet18 (middle) and Fisher vector representations (right), respectively. In each block, the leftmost column shows test examples, while the remaining five columns show the retrieved 5 nearest neigbors, sorted by the distance.

Figure S3: Left: Five nearest generated examples (column 2 to 5 in each row) of a given **training** example (column 1 in each row). Right: Five nearest training examples (column 2 to 5 in each row) of a given **generated** example (column 1 in each row). Both plots are generated with the Fisher Distance.

Figure S4: Generated samples in the early training phase (after 10K, 11K and 12K iterations, from left to right). Top: samples from using the default $G$ update. Bottom: samples generated using using the MCMC objective in Equation 7 with $\lambda = 1$. Note how with the MCMC objective, samples change gradually (compare same patch position across the three training checkpoints) and exhibit better quality and diversity at the same time, with fewer artifacts. (Best viewed in high resolution)