[Reviews · NeurIPS 2019]

Reviewer 1



Update: Author rebuttal promised to update the paper to address my criticism regarding clarity of methodological explanation. Contingent upon these changes to the camera ready, I have decided to increase my score from a 7 to an 8. This paper provides a view of generative adversarial networks through the lens of energy-based models (EBM). A slight modification to the original GAN objective falls out of this view, which provides a number of beneficial properties. Overall, I think this is a strong paper. It addresses numerous deficiencies in current GAN training methodology, namely the difficulty with evaluating the results and monitoring the stability of the training. The fact that the representations it learns are SOTA for unsupervised approaches on CIFAR10 is also impressive My main complaint is that, after reading, it is not entirely clear to me to what extent this strategy deviates from typical GAN training. Phrases like “we examine GANs through the lens of … “ (abstract) and “GANs can be reinterpeted” (conclusion) are used which seem to imply that the proposed approach does not alter ordinary GAN algorithms and is just a different perspective. However, the proposed approach *does* in fact alter the algorithm (line 72). More confusion is added when Equation 3 is introduced as an “optimization procedure”, when in fact it is merely a value function that does not immediately imply a procedure or algorithm. It would be great if the authors could clearly and concisely state how their method deviates from standard GAN training, as right now this information is scattered throughout the paper. This paper is also weakened a bit by the exclusion of code, the inclusion of which would greatly improve the accessibility of the work.

Reviewer 2



This paper continues along a thread in the literature linking GANs and deep energy-based models, the basic idea being that the discriminator can represent an energy function for the distribution and the generator a sampler for the same; this allows, among other things, a sampling approximation of the negative phase term (the gradient of the partition function) using samples from the generator. Taking this view, the manuscript under consideration proposes to leverage the gradient of the discriminator’s parameters to produce both Fisher vectors and a (diagonal approximation to the) Fisher information matrix for the model distribution. This allows for a powerful form of unsupervised representation learning, an induced distance metric (both between points and between sets of points, by applying the distance measure to the means of the sets). Overall, I feel this is a solid piece of generative model research. It proposes a fresh take on well-worn territory, makes several principled contributions as regards training methodology, and empirical demonstrate the method’s usefulness, in particular a classification result from unsupervised representation learning that is quite impressive. My criticisms below mainly concern the text. Obviously I’d really like to see how this stacks up on full-resolution ImageNet classification against recently proposed mutual information and BiGAN approaches, but I understand this could be resource-intensive and perhaps not feasible during the rebuttal period. One glaring omission from the discussion of related work is Ravuri et al’s method of learned moments work, which also uses the discriminator parameter gradient albeit for a different purpose. Lines 100-102 start off a sentence with “Intuitively” and then continue with an “if and only if” statement. Surely something more formal can be said, and this “intuition” can be made concrete. Footnote on page 3 refers to entropy regularization but AFAIK the entropy is not computed or approximated in the referenced work, rather “minibatch features” are computed as a way of detecting low diversity. Section 3.2 is a bit fuzzy on the details, and in particular the repeated references to MCMC are confusing. As best I can determine the loss term is MCMC *inspired*, but MCMC is not performed (otherwise lots of details are missing!). I’d try to make this clearer throughout. Also the use of the script I both for the Fisher Information and for the Identity matrix on 135 is potentially very confusing. The loss term proposed is quite similar in spirit to the historical averaging discussed in Salimans et al, except that it is realized in sample space rather than parameter space. It would probably be appropriate to make this connection explicit. 5.1 mentions a sigmoid nonlinearity (omitted from the referenced work since it was a WGAN) but then a least squares loss. Can you confirm, and clarify in the text, that it is a least squares loss applied to sigmoidally squashed outputs? This seems quite strange, though not unheard of (if I recall correctly, contractive autoencoders of Rifai et al used something similar). Probably appropriate to add Deep Infomax to the results table for CIFAR10 classification (to which you are superior). Re: “higher propensity to overfitting”: it’s not obvious to me that this should be the case, given that while the representations are higher dimensional they are also highly structured, as opposed to something like a feedforward autoencoder bottleneck. Post-rebuttal: I've reviewed the authors' responses, and am satisfied with their clarifications. While larger scale experiments would be ideal, I note and understand that there are difficulties inherent. I believe with the modifications and clarifications requested this is a very solid contribution, and have raised my score to an 8.

Reviewer 3



Thanks for the rebuttal. I have read it carefully, I stand by my original review and rating. The additional experimental results look good, and except for the scalability issue, most of my concerns are addressed. -------------------------------------------------------- The contributions are interesting and the paper is well written, the authors conduct extensive experiments on several tasks, including unsupervised feature extraction on CIFAR10 and CIFAR100, evaluated by a linear classifier, both the quantitative classification accuracy and the qualitative distance matrix demonstrates the AFV is an effective feature extractor despite at the cost of increased dimensionality of feature space. The paper also presents experimental results on using fisher distance to check the training process. It shows Fisher distance is a good substitute for objective metrics IS and FID when there is no classifier. Some detailed comments are listed below. - I would suggest adding some literature of fisher information and its use case in representation learning such as [1]. - Formula 4 and 5, why use samples from generator distribution to estimate the fisher score/information rather than real samples or combined? - One concern is the scalability issue, even with diagonalization the resulting fisher vector is still proportional to the number of parameters. In large scale GANs, such as [2], the method could be intractable. - It’s useful if the authors could list the performance of supervised training with the same architecture on CIFAR for comparison in table 1. - what type of gan loss function did the authors use in section 5.3? Besides varying the number of training examples, I think the authors should try different type of gans and check whether fisher distance is able to monitor the training dynamics for miss modes or gradient saturation. [1] Achille, Alessandro, et al. "Task2Vec: Task Embedding for Meta-Learning." arXiv preprint arXiv:1902.03545 (2019). [2] Brock, A., Donahue, J., & Simonyan, K. (2018). Large scale gan training for high fidelity natural image synthesis. arXiv preprint arXiv:1809.11096.

[Author Response · NeurIPS 2019]

We thank all the reviewers for their insightful comments. We will polish the wording accordingly and will add additional
references as suggested into the next version. We will also release a sample source code example demonstrating the
training and computation of AFVs. Below we address the major concerns, and for comments we do not respond to
explicitly we assume that we agree with the reviewer and will address properly in the revision.

1. Training protocol [**Reviewer #1**]: We apologize for the confusion in the training protocol. The simple answer is
that the EBM view does not necessarily change the way the model is trained. As a matter of fact, in the majority of
our experiments, we adopt the standard training procedure of GANs, except for cases where we explicitly test the
effectiveness of the MCMC-inspired objective (see Sec 5.4).

A more detailed explanation is as follows. First, we agree that Equation 3 is not a concrete optimization procedure.
However, it does indicate that, in order to train the EBM (D) with the variational trick, G needs to trained until
convergence to tighten the lower bound on the NLL before updating D. This is in contrast to what is suggested by
Equation 1 (following the convergence analysis of GAN theory), where D needs to be trained until convergence before
updating G. In practice, we approximate both of the max and min optimize problems in Equation 1&3 with a few steps
of mini-batch SGD, as is done in a standard GAN training procedure. As a result, the mini-batch SGD optimization
algorithm can be interpreted as an approximation to either of the two objectives. We will add proper clarifications to the
manuscript.

2. Scalability [**Reviewer #2,4**]: Scalability is a practical limitation of our work, because of two reasons. First, computing
Fisher Vectors amounts to taking the gradient of each example w.r.t. all model parameters, which is hard to parallelize
in modern deep learning frameworks. Second, the dimensionality of the induced Fisher Vectors is usually extremely
high, posing a heavy memory demand on GPUs. We made a few initial attempts, one of which is to modify the training
procedure such that we compute only the aggregated Fisher Vector representation for a mini-batch, and accordingly
modify the ground-truth to be the averaged labels of the batch. This resembles an extreme version of mixup, and works
reasonably well when batch size is small, but suffers from performance drop when batch size increases. We suspect that
smarter ways of constructing the batch and averaging the labels might lead to further improvements. For the second fact,
we also tried sampling the parameters to reduce the dimensionality of AFVs, but experienced minor performance drop
in classification accuracy. We believe that the full solution to the scalability issue deserves an independent contribution
and will leave it as future work.

3. Loss function [**Reviewer #2,4**]: Our default loss function is least square loss as in LSGAN, with sigmoid activation
on the output of D, except the experiment in Sec 5.4 where we explicitly test the MCMC objective in a new setting. The
reason for such a choice is that it provides the best numerical stability w.r.t. the outputs of D by preventing unnecessary
shifts of D's outputs. We have also tested the hinge loss as done in, e.g., BigGAN, which works equally well w.r.t. the
sampling quality and induced AFV representations, but weakened the smoothness of the Fisher Distance as a monitoring
metric. We will add proper clarifications and discussions.

4. The MCMC objective [**Reviewer #2**]: Your understanding is correct: MCMC is never actually performed. We refer
to MCMC because it offers an interpretation of the generator update as approximating one step MCMC. As a result, in
practice we can directly adopt the standard G update rule assuming that each G update is small, mimicking an MCMC
update. In cases where the local update of G is violated, it is useful to explicitly incorporate the proposed MCMC
inspired objective in Equation 7 as a regularizer, as shown in Sec. 5.4. We will make this clear in the paper.

5. Additional baselines for the classification experiment [**Reviewer #2,4**]: Per Rev 4's request, during the rebuttal
period we tested the supervised learning performance using the discriminator architecture, by changing the output
dimension of the last layer to 10. With only this change, the supervised learning test accuracy is 86.1%, which is worse
than our AFV + SVM's 89.1%. We then replaced all the Spectral Normalization layers with Batch Normalization and
repeated the experiment, and got a 92.7% accuracy, which exceeds our AFV result. We additionally have conducted the
same experiment on CIFAR100, where AFV+SVM achieves a test accuracy of 67.8%, compared to the supervised
training (with BN) performance 70.3%. Note that 67.8% is also the best result we can find under the pretraining + linear
classifier training setup on CIFAR100. For example, the Deep InfoMax paper reports an accuracy of 49.74%, which is
significantly worse than our result. We will report these experimental results in the paper.

6. Approximating Equation 4 [**Reviewer #4**]: The expectation term in Equation 4 is w.r.t. the model distribution $p_\theta$. It
is most natural to use the generated samples to approximate it because, according to the EBM view, the generator is
exactly trying to match the model distribution. Empirically, we also found that using the generated examples works
slightly better than using real examples, but the margin is small. We will clarify in the manuscript.

[Meta-Review · NeurIPS 2019]

The reviewers form a consensus that the paper is a nice paper!